# 3D Human Organoids: The Next “Viral” Model for the Molecular Basis of Infectious Diseases

**DOI:** 10.3390/biomedicines10071541

**Published:** 2022-06-28

**Authors:** Shirley Pei Shan Chia, Sharleen Li Ying Kong, Jeremy Kah Sheng Pang, Boon-Seng Soh

**Affiliations:** 1Disease Modeling and Therapeutics Laboratory, ASTAR Institute of Molecular and Cell Biology, Singapore 138673, Singapore; e0325971@u.nus.edu (S.P.S.C.); e0323545@u.nus.edu (S.L.Y.K.); pangks@imcb.a-star.edu.sg (J.K.S.P.); 2Department of Biological Sciences, National University of Singapore, 16 Science Drive 4, Singapore 117558, Singapore

**Keywords:** organoid, infectious diseases, viruses, bacteria, parasites, fungi, prions, pathogenesis

## Abstract

The COVID-19 pandemic has driven the scientific community to adopt an efficient and reliable model that could keep up with the infectious disease arms race. Coinciding with the pandemic, three dimensional (3D) human organoids technology has also gained traction in the field of infectious disease. An in vitro construct that can closely resemble the in vivo organ, organoid technology could bridge the gap between the traditional two-dimensional (2D) cell culture and animal models. By harnessing the multi-lineage characteristic of the organoid that allows for the recapitulation of the organotypic structure and functions, 3D human organoids have emerged as an essential tool in the field of infectious disease research. In this review, we will be providing a comparison between conventional systems and organoid models. We will also be highlighting how organoids played a role in modelling common infectious diseases and molecular mechanisms behind the pathogenesis of causative agents. Additionally, we present the limitations associated with the current organoid models and innovative strategies that could resolve these shortcomings.

## 1. Introduction

Claiming more than six million lives as of May 2022, the severe acute respiratory syndrome coronavirus 2 (SARS-CoV-2) has once again highlighted the deadliness of infectious disease [1]. At the same time, communicable diseases such as lower respiratory tract infections, diarrheal diseases, malaria, tuberculosis and human immunodeficiency virus (HIV)/acquired immunodeficiency syndrome (AIDS) remain the top 10 causes of global mortality [2]. The causative agents of infectious diseases are not confined to viruses, but also include bacteria, fungi, parasites and most recently proteins known as prions [3,4]. Hence, targeted preventive measures such as vaccines, and therapeutic interventions need to be developed to address this persistent and evolving threat brought about by these infectious pathogens. To win the infectious disease arms race, biologically relevant model systems that are efficient and robust in determining pathogenesis mechanisms are essential.

In recent years, 3D organoid models have been recognised as an up and coming in vitro construct, which can closely resemble the in vivo organ. In fact, the term “organoid” has existed since the 1940s, but its definition has changed over time [5]. Eventually, organoid was used to describe an in vitro 3D construct that is derived from human stem cells that are either pluripotent (embryonic or induced) or adult stem cells [5,6]. These stem cells are first differentiated to organ-specific cell types and self-organise into aggregates via cell sorting and spatially restricted lineage commitment [7,8]. Ideally, with the incorporation of multipotent cells and its ability to self-organise, an organoid would be able to recapitulate the architecture and functionality of the organ in an in vitro setting. As a more physiologically relevant model that maintains experimental tractability, the organoid system is an ideal method for the establishment of a more accurate mechanistic understanding of human infectious diseases. Several reviews have covered the formation of organoids in addition to utilising the organoids for modelling immunity and host–pathogen interactions with organoids [9,10,11,12]. Here, we present a comparison between the conventional models versus organoids while highlighting common infectious diseases that have been modelled using organoids, as well as future applications.

## 2. Comparison between Models of Infectious Diseases

Starting from the most simplistic model, immortalised cell lines such as HeLa cells have been widely used for biomedical research. Since the establishment of HeLa cells in the 1950s, prominent discoveries on infectious diseases, such as the route of entry for HIV and the pathogenesis of tuberculosis have been made [13,14,15]. Subsequently, epithelial, endothelial, immune and neural cell lines from different organs and tissues have been generated [16]. Aside from the extensive range of cell lines available, they are also affordable, indefinitely proliferative, homogenous and easy to culture [10,16]. These advantages allowed research to be conveniently performed with high efficiency. An example of an application would be in high-throughput screening for anti-virulence drugs [17]. Conversely, immortality renders these cell lines distinct from normal cells as they are either cancerous or genetically modified to divide infinitely [16]. Furthermore, after extensive rounds of passages, the cell line may undergo genetic alterations that change the phenotype, affecting the reliability of the results [16]. Hence, they may not be the best model to fully recapitulate the functional characteristics of the biological system, limiting the translational success rates from immortalised cell lines to humans.

The primary cell line is another 2D in vitro model that can better represent healthy normal cells. These cells are isolated from specific tissues of patients and are not subjected to genetic manipulation [16,18]. Hence, their attributes would show a higher resemblance to the original cell type in the in vivo condition. However, their implementation is often restricted by their lifespan in culture and oftentimes senescence already becomes prominent after the process of selection [16,18]. Since one of the hallmarks of aging is an increase in basal inflammation, infectious disease studies using this model might be capturing an inaccurate representation of the inflammatory response [19,20]. Furthermore, the in vitro conditions may not be optimal for cell survival, thus posing huge difficulties in culturing these cells [10,18]. 

Despite these drawbacks, these 2D monolayer models served as an affordable and simplified option in the advent of infectious disease study [21]. Nonetheless, the simplicity of these models still falls short of mimicking the complex microenvironment in which cells are situated, as they lack interactions between different cell types, especially the cells responsible for immune responses [10]. Therefore, to model this complexity and gain a more comprehensive understanding of infectious diseases, researchers often turn to in vivo modelling.

Even though costly to maintain, animal models are more physiologically relevant in terms of organismal complexity. On that account, model organisms are better at replicating the intercellular interactions and disease pathogenesis compared to monolayer cell culture. These in vivo systems have made significant contributions in identifying causative pathogens such as *Bacillus anthracis*, *Mycobacterium tuberculosis* and the rabies virus for their respective diseases [22]. By gaining a deeper insight into the molecular basis of the infectious disease, therapeutics such as vaccines and antimicrobials could then be developed. Subsequently, the safety and efficacy of these therapeutics could be evaluated via these animal models, bridging the gap from bench to bedside.

However, it is generally recognised that no model organism could fully mimic responses to infectious disease due to species-to-species variation. Infectious agents are species-specific and many have a limited host range [11,23,24]. Hence, not all in vivo models are compatible in the study of every infectious disease. Some pathogens are even known to only have a single host that they readily infect [25]. For example, HIV, human papilloma virus and measles virus are known to only infect humans [25]. Due to the inherent differences, not all findings yielded from animal studies could be directly translated to humans [11,23]. Specifically, immune system differences in commonly used murine models have restricted the transferability from murine to human [26,27]. One alternative is the humanised mouse model, but there are ethical concerns with excessive usage of animals [22,23,27]. With cost and ethical challenges, scalability is limited. Furthermore, pathogens could mutate and adapt to animal models. This is seen in the case of Ebola Zaire virus with emerging strains that are specific to mice and guinea pigs [28]. In the process of cell-culture, the virus may undergo mutations, which cause it to deviate from the clinically isolated form. This could then undermine the representation of the original pathogenesis. In the worst scenario, there is also a possible risk of zoonosis while working with animal models that could lead to an outbreak.

Considering the limitations in conventional models, a human pluripotent stem cell-derived organoid is a promising new in vitro technique that could address these gaps. Being an in vitro model, the 3D organoid technology is more scalable and cost-effective compared to animal models [11,29,30]. Other than the practical considerations, the properties of a 3D organoid supersede its 2D counterpart in terms of physiological relevance. Given that stem cells could differentiate into multi-lineages that are found within an organ, these self-aggregated 3D constructs could closely replicate the development and eventual composition of the organ, such as the presence of different cell types [8]. Hence, as compared with the 2D monotypic cellular model, the 3D organoid would better correspond to the in vivo equivalent in terms of structure and functions. Furthermore, since it originates from human stem cells, host–pathogen interactions that are specific to humans could be captured. Stem cells could be reprogrammed or directly isolated from patients to develop patient-derived organoids [31]. By doing so, the genomic and transcriptomic landscape of the patient would be preserved [31]. These organoids can be used for the investigation of heterogeneity in the individual host response to infection. Consequently, the results derived using organoids would be highly translatable to humans. Additionally, with a growing awareness of animal welfare, the scientific field is transiting away from the usage of in vivo model organisms [32]. Hence, human stem cell-derived 3D organoids could emerge as an excellent substitute, while providing ease of manipulation in in vitro settings.

Nevertheless, the organoid system has not completely replaced conventional research tools, as the simplicity of a 2D culture is still appreciated when it comes to pinpointing the cell population that is responsible for a certain effect. Due to heterogeneity present in organoid models, researchers would need to analyse these organoids in high resolution to identify cell-type-specific involvement [33]. This explains the complementary rise in single-cell based technologies [33]. In addition, the complexity of an animal model is still essential to capture the effects of immune system response, vascularisation and inter-organ interactions, all of which are limited in the organoid system [11]. Therefore, the utilisation of organoid technology is currently complementary to conventional methods.

A summary of the comparison between the different systems used in modelling infectious diseases is presented in Table 1.

## 3. Common Infectious Diseases Modelled by Organoids

Following its recognition as a compelling model, the scientific community has established organoids representing different in vivo tissues. Many have also harnessed the organoid technology for infectious disease modelling through the usage of brain, lung, gastric, intestinal, liver, kidney, heart and reproductive tract organoids (Figure 1). In this review, we will be examining some of the infectious diseases modelled using these tissue organoids. A summary of the key findings is presented in Table 2.

### 3.1. Brain Organoid

Brain organoids have been widely used as a model to study central nervous system infections. For instance, brain organoids were used to help elucidate the pathogenesis and infection of Zika virus (ZIKV) since they can recapitulate important phases of human foetal brain development. ZIKV infections were strongly linked to the upsurge of microencephaly and neurological complications in newborns [81,82]. Garcez, et al. [42] showed that ZIKV infection of human pluripotent stem cell-derived brain organoids resulted in stunted growth and provided evidence on how ZIKV infection results in severe damage in brain development. Similarly, another study showed that ZIKV-infected brain organoids exhibited disrupted cortical layers and a decline in proliferative zones, which further proves that ZIKV causes birth defects such as microencephaly [43]. Moreover, human cerebral organoids played an important role in delineating the pathogenesis of ZIKV. A study identified the association of ZIKV-mediated toll-like receptor 3 (TLR3) activation and dysregulation of neurogenesis together with organoid shrinkage, which is observed in microencephaly [44].

The human brain is also susceptible to prion diseases such as the most common sporadic Creutzfeldt–Jakob disease (CJD), a neurodegenerative disease that is deadly and transmissible. Due to the lack of a reproducible human cell model for prion infection, cerebral organoids have been adopted as a new model of human prion diseases to further investigate the infection, transmission and pathogenesis of prions in humans. Human cerebral organoids injected with two sporadic CJD prion subtypes exhibited uptake and clearance of the inoculated infectious agent, together with re-emergence of prion self-seeding activity and de novo propagation [34]. In the same study, brain organoids also showed varying human prion subtype pathologies. Moreover, the brain organoids had modifications in cellular metabolism and cytokine secretion such as an increase in chitinase 3-like-1 secretion, which has also been observed in the brains of deceased CJD patients. This phenomenon then contributes to neuronal and oligodendrocyte death [34]. Brain organoids were also used to model familial prion diseases. In a study, human cerebral organoids were generated using human pluripotent stem-cells derived from subjects harbouring the E200K mutation in the prion protein gene, the most common cause of human familial prion disease. However, the group showed that the 12-month-old cerebral organoids did not demonstrate the presence of insoluble, protease-resistant and seed-forming prion species, which indicates that the mutation alone is not disease-causing [35].

Severe acute respiratory syndrome coronavirus 2 (SARS-CoV-2) is a respiratory virus which also affects the central nervous system, resulting in several neurological complications and neuropsychiatric disorders [83,84]. Human pluripotent stem cell-derived brain organoids are greatly valued for the study of the molecular pathogenesis of COVID-19 in the central nervous system, given the scarcity of clinical brain samples of patients. Studies using brain organoid models showed neurotropism of SARS-CoV-2, whereby few neurons, astrocytes and glial cells were infected [38,39]. Choroid plexus epithelial cells were heavily infected in region-specific brain organoids, revealing how the virus contributes to the destruction of the human blood-cerebrospinal fluid barrier [36,37]. The molecular and cellular mechanisms in terms of host–virus interaction of SARS-CoV-2 were further investigated. Ramani, et al. [39] revealed that SARS-CoV-2-infected cortical neurons in human brain organoids showed mislocalisation of the microtubule-associated protein Tau from the axon to the soma, together with phosphorylation at threonine 231 of Tau. Neuronal cell death in SARS-CoV-2 was also observed in infected brain organoids, thus providing insights into the neurotoxic effects of SARS-CoV-2. In another study, Song, et al. [40] conducted single-cell RNA sequencing that revealed increased expression of genes related to metabolism and cellular reproduction in infected neuronal cells. On the other hand, uninfected neuronal cells showed enrichment of genes related to the hypoxic response pathway in the brain organoids. Song, et al. [40] then concluded that such a unique hypermetabolic state of SARS-CoV-2-infected cells allowed for efficient replication of the virus in neuronal cell types while inducing a hypoxic environment locally to inflict more damage to surrounding cells. In addition, Jacob, et al. [36] also showed transcriptional dysregulation in the choroid plexus organoids via single-cell RNA sequencing, whereby inflammatory cytokines were upregulated while the expression of transporters and ion channels essential for cerebrospinal fluid secretion was decreased. This led the team to conclude that SARS-CoV-2 infection of the choroid plexus cells triggered an inflammatory response that accompanied the disruption of the CSF-blood barrier function. SARS-CoV-2 infection of the brain organoid can be prevented by inhibiting ACE2 using antibodies or by supplying the organoid with cerebrospinal fluid from a COVID-19 patient that contains IgG antiviral antibodies [40]. Moreover, a Dicer isoform called antiviral Dicer (aviD) exhibited antiviral RNAi activity against SARS-CoV-2 in brain organoids by cleaving viral double-stranded RNA [41].

### 3.2. Respiratory Organoid

Riding on the wave of the COVID-19 pandemic, organoids have come under the spotlight and become a tool favoured by virologists. Considering that SARS-CoV-2 leads to severe pneumonia, the respiratory tract is the organ of concern [85]. To determine the tropism of SARS-CoV-2, Milewska, et al. [49] used a human airway epithelium (HAE) organoid constituted of multiple cell types, such as basal, ciliated and goblet cells. The diverse cell types within the organoid were able to mimic the mucosal barrier as well as surfactant proteins present within the in vivo microenvironment [49]. SARS-CoV-2 enters the host cell by binding to the angiotensin-converting enzyme 2 (ACE-2), followed by cleavage of the spike protein via transmembrane serine protease 2 (TMPRSS2) [86]. Several organoid studies have then revealed that primary infection occurs specifically in the ciliated cells of the airways as they constitutively express ACE2 [49,50,51]. Given the ability to capture the spatial orientation of the epithelium, the HAE organoid also demonstrated that the viral entry and release happens at the apical of the epithelium instead of the basal [49]. In another study, an HAE organoid had a viral replication profile and delayed interferon-stimulated gene (ISG) induction comparable to those observed in SARS-CoV-2-infected patients [52]. Due to this ability, the airway organoid model was able to demonstrate a higher viral entry rate for SARS-CoV-2 variants with a p.Leu452Arg mutation as compared to variants with a p.Asp614Gly mutation [87]. In addition, the former study also revealed that prior rhinovirus infection was able to hasten ISG induction in an organoid model, which can limit SARS-CoV-2 replication and reduce disease severity [52]. This receptor-mediated infection was further confirmed through lung alveolar organoids comprising mesenchymal cells, and both type 1 and 2 alveolar epithelial cells. Since the majority of the ACE2 receptors are expressed in type 2 alveolar cells, the organoid system revealed that these cells were particularly permissive to SARS-CoV-2 infection [88]. In another broncho-alveolar organoid model, the features presented due to targeted infection of type 2 alveolar cells in the organoid were comparable to the in vivo system [54]. Aside from mechanistic findings, it has been proposed by Bose [89] that lung organoids originating from varying ethnicities can be used to understand the infectivity of SARS-CoV-2 across the world. Further findings using organoids are also documented in numerous reviews [53,88,90,91,92,93].

Prior to the emergence of SARS-CoV-2, airway organoids were regularly utilised to study influenza viruses, mainly the influenza type A and B viruses. Virus tropism has been investigated for both types using the organoid model. An organoid model showed virus tropism and replication kinetics of influenza A virus, which were also observed in an ex vivo human bronchus explant [46]. It was also revealed that cell types such as ciliated cells and goblet cells were vulnerable to infection by different strains of influenza A virus [46]. Another study showed that the influenza B virus preferably infects the upper respiratory tract over the lower lung while suggesting that it has a replicative rate and tropism similar to influenza A [47]. Additionally, the usage of lung organoids also revealed that the replication of influenza A virus tends to form a specific foci, and is not randomly dispersed as shown by previous monoclonal cultures [94]. During infection of the lung organoid, IFN-related genes and proinflammatory genes were activated to mediate an immune response [94]. In addition, the infectivity of different influenza strains has been assessed using a human airway organoid [48]. The validity of the organoid model was corroborated with results gathered from four different influenza strains. Compared to the avian-infective H7N2 and swine H1N1 virus, higher replicative rates of the human-infective influenza strains, H1N1 and H7N9, were observed in the 3D human airway organoid [48]. Therefore, the virus titre and replicative rate within the 3D human airway organoid could serve as a basis to assess the infectivity of future influenza strains.

Other than viruses, bacteria such as *Mycobacterium tuberculosis* (TB) remain a key communicable respiratory threat, especially in developing countries [95]. TB is not the sole pathogen of concern given that pulmonary infections caused by non-tuberculous mycobacteria have also been increasing [96]. A notable member would be the *Mycobacterium abscessus* (Ab), as it is highly antibiotic-resistant and is linked to pulmonary diseases such as cystic fibrosis [97]. Currently, lung organoids are not widely used in mycobacteria research. However, with the ability to recapitulate the heterogeneous cellular composition along with spatial organisation present in the lung, lung organoids have an edge over 2D cultures. In a recent study by Iakobachvili, et al. [45], a broncho-alveolar organoid was utilised to decipher mechanisms involved in the early phase of mycobacteria infection. This was not possible in previous animal model studies, as the animals are not the primary hosts and could only partially present the pathological and clinical signs [98,99]. Coinciding with previous studies, the airway organoid model also demonstrated that TB has a low tropism for epithelial cells [100,101,102]. A distinguishing feature discovered using the airway organoid was that Ab tends to thrive better than TB in the airways. This provides further evidence that the alveolar macrophages provide a hospitable microenvironment for the survival of TB [45,102,103,104]. Differences in inflammatory responses were also captured wherein TB infection heightens the expression of cytokines and antimicrobial peptides, while Ab does not [45]. Impairment in mucin expression was also noted for both TB and Ab [45].

### 3.3. Gastrintestinal Organoid

The gastrointestinal (GI) tract, which consists of the stomach and intestines, is responsible for digestion and assimilation of nutrients. Since the mucosa layer of the human GI tract acts as the primary line of defence against pathogenic infections, modelling the GI tract would aid in the understanding of host–pathogen interactions [105]. GI organoids have bridged an important gap in understanding the pathogenesis of human-specific pathogens in infectious diseases, given the lack of animal or in vitro models. Such 3D organoids have provided insights into the underlying mechanisms of infection by various pathogens, such as receptors, mechanisms of entry, epithelial barrier dysfunction and human cellular responses to infection [11].

Intestinal organoids have advanced the field of infectious diseases by enabling the culture and study of viruses. Human noroviruses (HuNoV) are one of the top causes of acute gastroenteritis worldwide [106]. However, the lack of a reproducible in vitro culture system has hindered the study of the pathophysiology of human norovirus infection and how the virus interacts with the host. In a landmark study, Ettayebi, et al. [61] successfully generated a human pluripotent stem cell-derived enteroid, which is able to mimic the human intestinal epithelium, to allow for the culture of multiple human HuNoV strains. This study also revealed that certain HuNoV strains require bile to initiate replication [61]. Another prominent enteric virus is the human rotavirus which is another major cause of severe and fatal gastroenteritis in children five years of age and below [62]. Recent studies have shown that human intestinal organoids could successfully model rotavirus infection using patient-derived strains [62,63]. Such human intestinal organoids have allowed for the study of the interaction between rotavirus and the human host. Yin, et al. [62] have shown that human intestinal organoids infected with rotavirus exhibited an activation of antiviral interferon (IFN) signalling, whereby various ISGs were highly expressed. With the elucidation of active virus–host interactions in the human intestinal organoids, the group further established that antiviral drugs such as interferon-alpha (IFN-α) and ribavirin could inhibit the replication of clinical rotavirus strains.

Recently, human intestinal organoids were discovered to be permissive to SARS-CoV-2 infection and replication which facilitated the study of SARS-CoV-2 tropism in various intestinal cell types [64,65,66,67]. Moreover, human intestinal organoids have shed light on the molecular mechanisms of intestinal SARS-CoV-2 infection. SARS-CoV-2-infected intestinal organoids exhibited induction of type III IFN together with ISGs and inflammatory cytokines [64,67]. Moreover, a similar inflammatory response involving the upregulation of IFN-related genes was exhibited in multiple epithelial cell types in human intestinal organoids [65].

Aside from viral infectious diseases, an intestinal organoid has also been used to model parasitic infection caused by Cryptosporidium. With Cryptosporidium being one of the main causes of diarrhoea worldwide, there is a need to understand its pathophysiology to develop therapeutics for it [107]. Previously, there was a lack of appropriate in vitro models that could support the life cycle of the parasite, as it requires a suitable host to thrive [108]. An intestinal organoid, on the other hand, was able to sustain the complete life cycle of Cryptosporidium and generate infectious oocytes that appear analogous to those generated by a host animal [60]. This implies that the intestinal organoid was able to simulate in vivo host conditions to allow the survival of the parasite. Furthermore, by utilising the fact that the intestinal organoid could be composed of either progenitor cells or differentiated cells, the tendency to infect differentiated cells was also identified [60]. Future studies could then investigate the specific receptors on differentiated cells that allow the preferential infection of differentiated cells.

Considering the similarities between the stomach and intestines, a gastric organoid was subsequently derived by modifying the protocol for intestinal organoids [109,110,111]. The establishment of a gastric 3D organoid facilitated breakthroughs for several infectious diseases, which previously lacked platforms to optimally study their pathogenesis. One prominent example is the *Helicobacter pylori* bacteria that afflicts more than 50 percent of the world’s population [112,113]. Considering that *H. pylori* infection is a major risk factor for chronic gastritis, gastroduodenal ulceration and gastric cancer, gastric organoids carry a great potential to recapitulate clinical manifestations [114]. It is known through gastric cancer cell lines that this bacterium activates nuclear factor kappa B (NF-κB) via the cytotoxicity-associated gene pathogenicity island (cagPAI) of the bacterium [115,116]. Subsequently, this triggers the downstream target gene, interleukin-8 (IL-8). On the contrary, the organoid system revealed that cagPAI of the bacterium is not responsible for NF-κB activation [55]. Even though the actual activator remains unknown, cell lineage may be a contributing factor [55,117]. This is because the study revealed that the cell lineage of the pit expressed a lower level of IL-8 compared to the cell lineage belonging to the gland, which suggested that the gland lineage mounts a strong inflammatory response [55]. This may be attributed to the presence of pit mucus, which serves as a barrier, and there is a possibility that gland-specific receptors could be involved in mounting an inflammatory response [55]. This finding was only possible due to the ability to simultaneously infect both lineages in an organoid. Furthermore, the ability to simulate the protective mucus barrier over the epithelial cells is an advantage of organoids over the conventional in vitro models [56,118,119,120].

The gastric organoid has also unveiled many underlying mechanisms of *H. pylori*, from receptor recognition, attachment, replication and disease progression. The presence of the mucus layer within the organoid demonstrated the colonisation of the mucosa by the bacterium upon exposure [55,57,58,59]. The CagA-ASPP2 complex was found to mediate the colonisation of *H. pylori* in the gastric organoid model [121]. The translocation of bacterial protein CagA into host cells and the subsequent interaction of the protein and tyrosine kinase receptor c-MET resulted in changes in cell morphology and a rise in cellular proliferation, which could be the cause of carcinogenesis [57,59,122,123]. Alteration of the state of differentiation of the gastric progenitor cells by *H. pylori* could also contribute to the fate of cancerous cells [124]. In another instance, organoids revealed that the activation of NF-κB by *H. pylori* resulted in the acute induction of Sonic Hedgehog (Shh) from the parietal cells in the stomach, which could contribute to gastritis [58].

### 3.4. Liver Organoid

Currently, liver organoids are able to carry out liver-specific functions such as protein synthesis and drug metabolism [125]. Additionally, vascularisation was also demonstrated upon transplantation of organoids into mice [125]. Moreover, these liver organoids contain functional hepatocytes and are genome-stable with few single nucleotide polymorphisms [126]. Due to these qualities, the liver organoid serves as a robust model to elucidate the pathogenesis of viral diseases such as chronic viral hepatitis. Using hepatoma organoids, a study showed that the hepatitis C virus (HCV) uses entry factors such as scavenger receptor BI (SR-B1), epidermal growth factor receptor (EGFR), cluster of differentiation 81 (CD81), claudin-1 (CLDN1) and occludin (OCLN) in a sequential actin-dependent manner, to facilitate entry into the organoid system [71]. Recently, organoids generated from liver stem cells of HCV-infected individuals showed viral replication and sustained a low-grade infection for months [72]. Moreover, single-cell RNA sequencing performed during the same study demonstrated that infected cells showed extensive transcriptional reprogramming, whereby cancer stem cell development, viral replication and hepatocyte differentiation were promoted, whereas IFN signalling and proliferation were antagonised [72]. Thus, this has shone light on how HCV infection might cause sustained liver damage and increase the risk of cancer in chronically infected patients [72]. Lately, liver organoids were also utilised to model hepatitis B virus (HBV) infection and these functional organoids were more susceptible to infection as compared to iPSC-derived hepatocyte-like cells [68]. HBV-infected liver organoids demonstrated hepatic dysfunction, accompanied by decreased expression of hepatic genes, increased release of prognostic markers for acute liver failure and altered ultrastructure [68]. To further mimic the presence of an immune response in the organoid, a multi-cellular system consisting of Kupffer cells, hepatocytes, biliary cells and stellate cells was generated [69]. This system would allow for more accurate modelling of processes such as inflammation and fibrosis that occur in infectious diseases [69,70].

Additionally, liver organoids provide a more feasible avenue to investigate pathogens with long dormant periods or which are prone to possible relapses. This is because, in contrast to the maximum viable culturing time of two weeks for 2D primary human hepatocytes, the liver organoid can remain viable and phenotypically stable in culture from 5 to 10 weeks. Moreover, a foetal tissue-derived liver organoid was able to be sustained for at least 11 months. This feature of prolonged survival is favourable for the study of malaria, where the causative Plasmodium protozoan parasites are harboured in a hepatocyte during the pre-erythrocytic phase of the life cycle, also termed the liver stage, for weeks or months [74]. This was evident in a study conducted by Arez, et al. [73], where the liver organoids were not only susceptible to *Plasmodium berghei* (murine-infective parasites) infection, but also supported the liver-stage infection, which eventually formed exoerythrocytic merozoites that are responsible for blood infection [73]. Furthermore, the 3D structure of the organoid also demonstrated that the *Plasmodium* sporozoites were able to cross the extracellular matrix of the organoid to infect the core [73]. This was not observable in 2D cultures as the infectivity is generally lower [73]. With the ability to recapitulate prolonged periods of infection, the usage of organoids could facilitate extensive research on malaria and other infectious diseases that require a long period of incubation.

### 3.5. Heart, Reproductive Tract and Skin Organoids

With recent studies showing that patients infected with SARS-CoV-2 develop cardiovascular complications and post-acute cardiovascular manifestations, cardiac organoids have been adopted to investigate the mechanisms in which the virus elicits cardiac dysfunction and to identify potential cardioprotective drugs [75,76]. It was revealed that human cardiac organoids exposed to an inflammatory “cytokine-storm” cocktail consisting of IL-1β, IFN-γ and poly(I:C) exhibited diastolic dysfunction [75]. The team went on to conduct phosophoproteomics on the cytokine cocktail-treated cardiac organoids, which showed enhanced phosphorylation of the signal transducer and activator of transcription 1 (STAT1) and two sites on the bromodomain-containing protein 4 (BRD4) [75]. Moreover, single nucleic RNA sequencing revealed that the human cardiac organoids also showed enhanced viral responses, which could be mediated by STAT1 and epigenetic activation including BRD4, and a similar response was also observed in SARS-CoV-2-infected K18-hACE2 transgenic mice [75]. With an enhanced mechanistic insight into how the virus induces cardiac damage, it was further shown that bromodomain and extraterminal family inhibitors (BETi) could ameliorate cardiac dysfunction in the organoids while decreasing viral responses transcriptionally and decreasing the infection rate of cardiomyocytes by SARS-CoV-2 [75]. With emerging infectious diseases, cardiac organoids could also be valuable in studying myocarditis due to infection by viruses and protozoa such as *Trypanosoma cruzi* (*T. cruzi*) [127,128]. *T. cruzi* infection results in Chagas disease, which is the leading cause of myocarditis [129]. Even though other in vitro models such as immortalised cell lines, primary cell lines and hiPSC-derived cardiomyocytes have been used, there have been outstanding questions, such as the different manifestations and time of onset of Chagas disease myopathy in patients [130]. In this context, patient-derived cardiac organoids could provide new insight into the pathophysiology of Chagas disease, such as the mechanism underlying *T.cruzi* persistence.

Due to limited access to reproductive material, there has been a lack of experimental models for the study of sexually transmitted diseases [131]. Organoids have therefore been widely used to study reproductive infectious diseases due to their ability to recapitulate the complexity of the in vivo features of the reproductive organs [132]. Human fallopian tube organoids were used for the long-term study of *Chlamydia trachomatis* (*Ctr*) serovars D, K and E infection [77]. From the study, it was demonstrated that the epithelium of the infected organoid expels *Ctr* bacteria into the lumen and undergoes compensatory cellular proliferation. Moreover, there was an activation of LIF signalling, which regulates stemness, in the organoids, which provides insight into how *Ctr* bacteria could initiate high-grade serous ovarian cancer. Recently, organoid-based cervical models have been adopted for the study of human papillomavirus (HPV) and how it promotes carcinogenesis in the cervix, resulting in cervical cancer [78]. It is evident that such reproductive organoids are highly valuable in vitro models for the study of human reproductive infectious diseases, enabling researchers to further investigate disease pathogenesis and the application of drug screening. Other sexually transmitted diseases caused by other pathogens such as *Trichomonas vaginalis*, *Neisseria gonorrhoeae* and herpes simplex virus could be further studied using organoid technology.

Human skin organoids, which are able to mirror the complex features of the human epidermis and remain expandable for half a year, have been developed for use in dermatological research. Recently, a human primary epidermal organoid system was found to be permissive to fungal infection caused by *Trichophyton rubrum* (*T. rubrum*) [80]. Through the study, it was revealed that the persistent skin infections coupled with minor inflammation were likely due to a constant suppression of interleukin (IL)-1 signalling [80]. The establishment of a functional skin organoid could drive future studies into the causative agents underlying skin infections such as the *Candida* species, *Streptococcus* and *Staphylococcus aureus* [133,134]. In a recent study, a skin organoid was also used to demonstrate that SARS-CoV-2 was capable of infecting both the hair follicles and the nervous system associated with the skin. Consequently, these provided evidence that COVID-19 patients are susceptible to hair loss and cutaneous lesions.

## 4. Limitations & Future Perspectives of 3D Human Organoid

As seen from the vast array of applications of 3D human organoids in infectious disease research, the organotypic model has contributed significantly to the advancement of infectious disease research. With both the ease of in vitro manipulation and physiological relevance similar to the in vivo system, organoids provide a simple and efficient method to investigate host–pathogen interactions and disease pathogeneses. Despite their promise in uncovering the molecular basis underlying infectious diseases, there is still room for improvement to enhance the applicability of these models.

One limitation is their lack of reproducibility [135,136,137]. This is because organoid cultures are grown on an extracellular matrix (ECM) hydrogel to provide structural support and to simulate the biochemical signals that are essential for the development of organotypic features [138,139,140]. This scaffold-based method allows for the long-term culture of the organoids [138,140]. However, these ECM hydrogels are often obtained by decellularisation of animal-derived matrices, which in turn means their composition is poorly defined [141,142,143]. Within the ECM, residual artefacts such as inflammatory proteins could then significantly limit the reproducibility of organoid formation [135]. To reduce this variability, synthetic matrices (e.g., alginate, hyaluronic acid, polylactic-co-glycolic-acid, polyethylene glycol and nanocellulose) that can be chemically defined are currently being explored [135,143]. In addition to the use of scaffold-based approaches, scaffold-free methods are also available. These methods involve the self-aggregation of a heterogeneous pool of cells in suspension to form tissues with endogenous ECM [140,144]. Taking into account the absence of a scaffold, rapid formation is possible, which can contribute to a higher throughput [140,144]. Further to these factors, genetic aberrations and aging that occur with prolonged culture of human pluripotent stem cells could also result in batch-to-batch variability during organoid culture [145]. Hence, further improvement in the protocol for pluripotent stem cell maintenance will be needed to minimise cellular aberrations. Furthermore, the limited period during which an organoid can be maintained in culture could restrict the maturity of the organoid [146]. Some studies have reported that organoids tend to retain a foetal phenotype, which could have hindered the representativeness of experimental findings that require a mature phenotype [147,148]. This varying degree of organoid maturation would also introduce heterogeneity in the endpoint results. Thus, refinement in terms of morphogenic control would be needed to ensure the consistent maturation of organoids. Alternatively, another method to improve reproducibility would be the application of bio-printing [149]. By standardising the formation of organoids, the variability in the size of organoids could be resolved [150,151]. Moreover, the yield of organoids generated through bio-printing would triumph over the scale brought about by traditional culture methods [150,151]. Although bio-printing could accurately control the dissemination of cells, the limited resolution of bioink still restricts its ability to give rise to microscopic vasculature such as capillaries [149]. With a higher resolution, the bioink tends to be of a lower viscosity, which impedes the structural assembly of macroscopic tissues [152]. A higher resolution would also cause greater mechanical stress to the cells during deposition, which in turn reduces cell viability [149,153]. Therefore, breakthroughs in organoid bioprinting would be required to push this technology to greater heights.

Additionally, a more comprehensive analysis of pathogeneses would require a greater complexity of the organoid systems. Importantly, infectious disease progression involves various immune responses and one key component lacking in most organoids is immune cells. To address this, researchers have started to establish co-culture models comprising both the organoid and immune cells. An overview of these co-culture models is noted in the review by Bar-Ephraim, et al. [154]. Even though these models are still in their infancy, they have the potential to serve as a critical tool in understanding immune responses to infectious diseases.

Apart from immune cells, these in vitro models also lack vascularisation networks, which are important for material exchange. Without vascularisation, the microenvironment encapsulated in the organoid may not be the best representation [155,156]. A necrotic core may form when diffusion of essential nutrients needed for survival is impeded as the organoid increases in size, hence limiting the scale of organoid models [8,155]. Currently, a range of vascularisation techniques for organoids are being explored. They can be separated into two categories: in vitro and in vivo vascularisation [157]. In vitro strategies involve a co-culture system with endothelial cells or the introduction of angiogenesis-inducing factors [158]. Other methods include the layer-by-layer deposition of cell-containing matrices via bio-printing, as well as the formation of tubular channels through the dissolution of sacrificial material or moulding of hydrogel constructs [158]. Conversely, these methods have been reported to disrupt the structure during the self-organisation of the organoid. Thus far, in vivo vascularisation established via the implantation of organoids within a host has been the most successful in developing a fully functional vasculature [156]. Nonetheless, this solution would then reduce the cost-effectiveness of the organoid system due to the need for an additional host.

Additionally, even though the organoid model could represent a single tissue or organ more comprehensively than a 2D culture, the pathological effects of infectious diseases are rarely localised in one organ. This is evident from the effects of SARS-CoV-2 on different organoids that we have presented in this review. Evidently, human organoids lack the ability to capture inter-organ interactions, which is crucial in understanding the systemic effects of various infectious diseases. Moving forward, a platform that allow inter-organ interactions to be examined will be desirable in the field of infectious disease research.

An example would be the incorporation of organ-on-chip (OoC) technology. The OoC technology involves the use of microfluidic cell culture devices that are able to recapitulate the tissue and organ level of function in a tightly regulated manner in vitro [159,160]. This technology has been able to fill in the gap of critical functions that 3D organoids have lacked, such as the presence of different tissue interfaces and barriers, spatiotemporal chemical gradients, recruitment of circulating immune cells, incorporation of mechanical cues and fluid shear stress, as well as the complex microbiome of the human body [161,162]. Moreover, there has been an advancement of body-on-a-chip platforms, whereby multiple organ-on-chip models are connected fluidically to mimic cross-organ communication, and such technology allows for modelling of systemic physiological responses [162,163]. The use of OoC technology in infectious disease research has allowed for a more accurate model of host–pathogen interactions, allowing for further understandings of the pathophysiology of infectious microbes [164]. Recently, organ chips were also used to model SARS-CoV-2 infection of the lung, in order to elucidate the mechanisms of its pathogenesis. Zhang, et al. [165] demonstrated that the human alveolar chip was able to mirror human SARS-CoV-2 induced lung injury and immune responses such as release of inflammatory cytokines and immune cell recruitment. Evidently, organ chips have various advantages over existing in vitro models and animal models as a valuable tool for infectious disease research. However, a combination of organoid and OoC approaches via synergistic engineering could bring about a more powerful, controllable and accessible in vitro system [159,166].

A summary of the discussed in vitro platforms can be found in Figure 2.

## 5. Conclusions

Human pluripotent stem cell-derived organoids have been monumental in the advancement of infectious disease research. Organoids are able to recapitulate native tissues in a tractable manner while resolving some of the limitations present in former conventional methods. This indicates that the organoid model is a valuable system to study the molecular basis of pathogens to combat the rising threat of infectious diseases. Despite its promise, shortcomings are also present, such as the lack of reproducibility and complexity that are essential to fully replicate the in vivo settings. This calls for standardisation of the organoid generation protocol as well as the incorporation of immune cells, vascularisation and inter-organ interactions through a micro-fluidics system as shown in next generation OoC technology. However, these technologies are still in the stages of infancy; thus, a combination of 2D, 3D and humanised animal models would provide a synergistic approach to uncover the mechanistic routes of infections of interest.

## Figures and Tables

**Figure 1 biomedicines-10-01541-f001:**
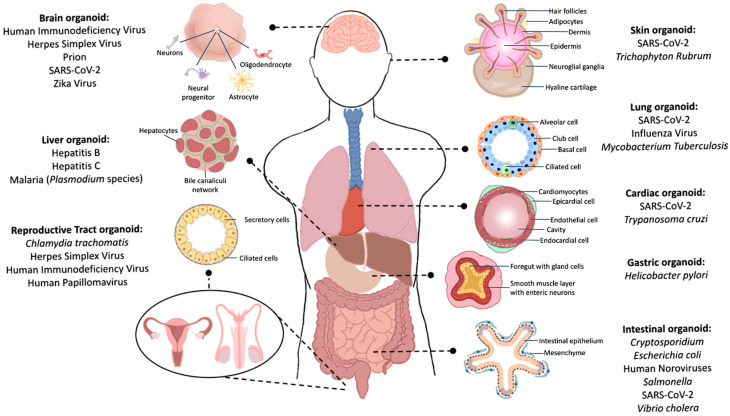
Common infectious diseases modelled using 3D organoids derived from human stem cells.

**Figure 2 biomedicines-10-01541-f002:**
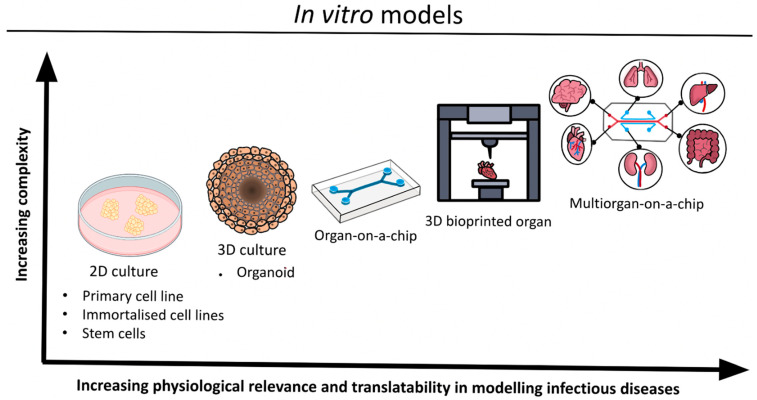
Schematic of in vitro platforms to model infectious diseases.

**Table 1 biomedicines-10-01541-t001:** Comparison between in vitro and in vivo systems.

Properties	In Vitro			In Vivo
	Immortalised Cell Line	Primary Cell	3D Organoids	Animal Models
**Technical Aspects**				
Cost	Low	Low to Moderate	Moderate	High
Ease of Handling	High	Moderate to High	Moderate	Low
Scalability	High	Moderate	Moderate to High	Low
Reproducibility	High	Low	Low to Moderate	Low
**Biological Aspects**				
Immune Response	No	No	No	Yes
Vascularisation	No	No	Yes (In some)	Yes
Physiological Relevance	Low	Moderate	Moderate to High	Moderate to High
Heterogeneity	No	No	Yes	Yes
Translatability	Low	Low	High	Moderate to High

**Table 2 biomedicines-10-01541-t002:** Summary of the key findings of the molecular basis of infectious disease using the organoid model.

Organ Modelled by Organoid	Infectious Pathogen	Molecular Basis					Reference
		Mechanism of Entry	Tissue Tropism	Replication/Propagation	Immune Response	Disease State	
**Brain**	Prion	√	√	√	√		Groveman, et al. [34], Foliaki, et al. [35]
	SARS-CoV-2	√	√	√	√	√	Jacob, et al. [36], Pellegrini, et al. [37], Ramani, et al. [38], Ramani, et al. [39], Song, et al. [40], Poirier, et al. [41]
	Zika Virus				√	√	Garcez, et al. [42], Cugola, et al. [43], Dang, et al. [44]
**Respiratory Tract**	*Mycobacterium* species		√	√	√		Iakobachvili, et al. [45]
	Influenza Virus		√	√			Hui, et al. [46], Bui, et al. [47], Zhou, et al. [48]
	SARS-CoV-2	√	√	√	√		Milewska, et al. [49], Hou, et al. [50], Hikmet, et al. [51], Cheemarla, et al. [52], Han, et al. [53], Lamers, et al. [54]
**Stomach**	*Helicobacter pylori*	√	√	√			Bartfeld, et al. [55], Jeong, et al. [56], Bertaux-Skeirik, et al. [57], Schumacher, et al. [58], Wroblewski, et al. [59]
**Intestines**	Cryptosporidium		√	√			Heo, et al. [60]
	Human Noroviruses	√		√			Ettayebi, et al. [61]
	Human Rotaviruses				√		Yin, et al. [62], Finkbeiner, et al. [63]
	SARS-CoV-2	√	√		√		Lamers, et al. [64], Mithal, et al. [65], Stanifer, et al. [66], Zhou, et al. [67]
**Liver**	Hepatitis B	√			√		Nie, et al. [68], Ouchi, et al. [69], Cao, et al. [70]
	Hepatitis C	√		√			Baktash, et al. [71], Meyers, et al. [72]
	Malaria (*Plasmodium* species)		√	√			Arez, et al. [73], Mo and McGugan [74]
**Heart**	SARS-CoV-2				√		Mills, et al. [75], Xie, et al. [76]
**Reproductive Tract**	*Chlamydia trachomatis*		√	√			Kessler, et al. [77]
	Human Papillomavirus					√	Lõhmussaar, et al. [78]
**Skin**	SARS-CoV-2		√				Ma, et al. [79]
	*Trichophyton rubrum*				√	√	Wang, et al. [80]

## Data Availability

Not applicable.

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
