# Peer review of "3D Human Organoids: The Next “Viral” Model for the Molecular Basis of Infectious Diseases"

_biomedicines, 2022, doi:10.3390/biomedicines10071541_

Round 1

Reviewer 1 Report

The review article entitled “3D Human Organoids: the next "viral" model for Molecular Basis of Infectious Diseases” provides information about the importance of the 3D Organoids model in studying Disease models and their molecular mechanisms. The manuscript lacks a few important topics such as Methods using organoids to study infections, Bioreactors for 3D Organoids, etc. Thus, issues are needed to be addressed first before the recommendation of this review article for publication

1.        Patient-derived organoid culture is an important source in studying various conditions like cancer, infection etc. Add a note on patient-derived organoid cultures and their importance in studying Infectious Diseases

2.        Describe the various techniques used for the development and culturing of the Organoids like suspension cultures, bioreactors etc.

3.        Add a section on 3D scaffold-free engineered organoids and their significance compared to scaffold-based organoid

4.        The author should cite the recent important reference in this review article. such as

https://doi.org/10.1128/jvi.00098-22

https://doi.org/10.1038/s41592-022-01453-y

https://doi.org/10.1007/s12015-020-09989-2

5.        Typographical errors need to be corrected throughout the manuscript.

Reviewer 2 Report

The present manuscript is quite insightful. However, this manuscript needs some minor changes before its consideration for the publication.

Comments:

1. References of Table 2 are missing from the manuscript. Reviewer suggests adjusting the table format according to the page size or vice versa.

2. Reviewer suggests incorporating one or two figures representing different types of organoids (their organization and functionality) to make this manuscript more appetizing and informative from figures.
